# Therapeutic DNA vaccine attenuates itching and allergic inflammation in mice with established biting midge allergy

**Mey Fann Lee[1]☉, Yi-Hsing Chen [ORCID]2,3,4☉*, Pei-Pong Song[1], Tzu-Mei Lin[2]**

**1** Department of Medical Research, Taichung Veterans General Hospital, Taichung, Taiwan, **2** Division of Allergy, Immunology and Rheumatology, Taichung Veterans General Hospital, Taichung, Taiwan, **3** Department of Life Science, Tunghai University, Taichung, Taiwan, **4** Faculty of Medicine, National Yang-Ming University, Taipei, Taiwan

☉ These authors contributed equally to this work.
\* ysanne@vghtc.gov.tw

**Data Availability Statement:** All relevant data are within the manuscript and its Supporting Information files.

**Funding:** This study was supported by grants from the Taiwan Ministry of Science and Technology

## Abstract

*Forcipomyia taiwana* is a tiny hematophagous midge that attacks en masse. It is responsible for the most prevalent biting insect allergy in Taiwan. For t 2 is its major allergen. The intense itchy reactions can prevent allergic individuals from performing their regular daily outdoor activities. This study aimed to investigate whether the For t 2 DNA vaccine was effective in treating mice with established biting midge allergy. Mice were sensitized with recombinant For t 2 proteins or whole midge extracts. Two to four consecutive shots of various concentrations of For t 2 DNA vaccine, with or without CpG adjuvants, were then administered to midge-sensitized mice. Mice that received two shots of 50–100 µg For t 2 DNA vaccine showed a significant reduction in allergen-induced bouts of scratching, For t 2-specific IgE, specific IgG1/IgG2a ratio in sera, skin eosinophil infiltration, and IL-31 production, as well as IL-4 and IL-13 production by splenocytes. Two doses of For t 2 DNA vaccine one week apart was sufficient to treat mice with established biting midge allergy. The treatment resulted in clinical, immunological, and histopathological improvements. We recommend that this low-cost, convenient treatment strategy be developed for use in humans who are allergic to biting midges.

## Introduction

The biting midge, *Forcipomyia taiwana*, is a very small (1–1.5 mm) blood-sucking insect, which is widely distributed throughout Taiwan and southern China [1]. It is the most prevalent biting insect allergy in Taiwan, and as many as 60% of exposed individuals develop intense itchy reactions to the bites; 58–64% of individuals are sensitized to its major allergen, For t 2 [2–4]. The hematophagous genus of *Forcipomyia* has been reported around the world, including Singapore, Poland, Italy, North India and Mexico [5–9]. Unlike mosquitos or other midges, *F. taiwana* do not transmit infectious diseases to human subjects through biting and blood sucking [10], however they can cause annoying, intense, itchy allergic reactions. People

(grant nos. MOST-105-2321-B-010-017 and MOST-106-2321-B-010-013). The URL of the funder is www.most.gov.tw. The funder did not participate the study design, data collection and analysis, decision to publish and preparation of the manuscript.

**Competing interests:** The authors have declared that no competing interests exist.

with allergic reactions to insects frequently avoid normal outdoor activities, which may negatively impact their quality of life [11–13]. Allergic reactions to the bites of members of the midge family are not isolated to humans, they also occur in other animals, such as horses [14].

The major allergen of *F. taiwana* is For t 2; it has a 65–77% overlap with the eukaryotic translation initiation factor 3 subunit of many insects. The For t 2 protein not only binds with serum IgE in patients allergic to midges, it also induces the production of key inflammatory chemokines from skin fibroblasts in a concentration-dependent manner [3].

A previous study by the authors showed that a DNA vaccine encoding the For t 2 midge allergen was able to prevent the development of allergic skin inflammation induced by the biting midge allergen in a mouse model [15,16]. However, as biting midge allergy is so prevalent in Taiwan, a large scale preventive vaccination does not seem to be practical in the real world. A vaccine that is able to treat patients with an established midge allergy is more clinically relevant.

The present study aimed to investigate whether the For t 2 DNA vaccine was effective in treating mice with an established biting midge allergy.

## Materials and methods

### Expression and purification of the recombinant For t 2 from *E. coli* for sensitization

Plasmid pQE30 containing For t 2 coding sequences was transformed into *Escherichia coli* strain M15, the protein expression was performed as previously described [3]. After a His-tag affinity column, the *E.coli*-expressed For t 2 recombinant protein (*E*-rFor t 2) was further purified using Endotoxin Detoxi-Gel (Pierce, Rockford, IL, USA) and passed through a 0.22 μm syringe filter (Millipore, Billerica, MA, USA) as previously described [15]. The endotoxin content was determined using an E-TOXATE kit (Sigma-Aldrich; Merck KGaA, Darmstadt, Germany). The lowest detection limit of the test was 0.05 endotoxin units per ml.

### Preparation of pCI-For t 2 plasmid for vaccination

The eukaryotic expression plasmid pCI-For t 2 was propagated in *E. coli* DH5α and large-scale purification was performed with the EasyPrep EndoFree Maxi Plasmid Extraction kit according to the manufacturer's instructions (Tools, New Taipei City, Taiwan). The expression level of pCI-For t 2 was confirmed in mammalian cells by transfecting the human epidermal keratinocyte cell line HaCaT (ATCC, Manassas, VA, USA) with 2.5 μg of the plasmid using Lipofectamine 2000® (Invitrogen: Thermo Fisher Scientific, Inc., Waltham, MA, USA). The cells were cultured for 24 h and then the supernatants from the transfected cells were examined by western blot analysis using rabbit anti-*E*-rFor t 2 polyclonal antibodies.

### Experimental design of therapeutic DNA vaccination

Female 6-week-old BALB/c mice were purchased from the National Laboratory Animal Center, Taiwan and raised under specific pathogen-free conditions. All animal experiments were reviewed and approved by the Institutional Animal Care and Use Committee of Taichung Veterans General Hospital (approval no. LA1051380).

A total of 30 mice were sensitized with 2 intraperitoneal injections (IP) of 10 μg *E*-rFor t 2 absorbed by 2 mg alum adjuvant, with a 1-week interval between each injection. On days 14 and 21 the three groups of vaccinated mice (V50, V100 and V200) received intra-muscular (IM) injections of the For t 2 DNA vaccine at doses of 50, 100 and 200 μg, respectively. The not-vaccinated and vector only (VO) groups received IM injections of PBS and 100 μg pCI-

neo plasmid DNA as the controls. All groups of mice were challenged intra-dermally (ID) with 3 doses of 1 µg *E*-rFor t 2 for 3 consecutive days between days 59–61. Blood samples were collected bi-weekly from the retro-orbital venous plexus. The scratching behaviors of the mice were video recorded on days 0 and 61, and the mice were sacrificed on day 63.

Same protocol was performed for the second set of experiments using mice sensitized with crude midge extracts as well as experiments testing effects of CpG adjuvants.

## ELISA for the measurement of For t 2-specific antibodies in the sera

The titer of anti-For t 2 IgE, IgG1 and IgG2a antibodies in the mice sera was detected using ELISA. 96-well ELISA plates (NUNC MaxiSorp, Thermo Fisher Scientific, Inc.) were coated with 0.5 µg *E*-rFor t 2/well in a coating buffer and ELISA was performed as previously described [15].

## Evaluation of mouse scratching behavior immediately following an intra-dermal challenge

On days 0 and 61, the scratching behaviors of the mice were video recorded for 1 h starting immediately after the intra-dermal challenge with *E*-rFor t 2. Counts of scratching were made using video playback. The observation of scratching behavior was blinding measured and performed as previously described [15,16].

## Histological examination of delayed reactions

On day 63 the mice were sacrificed and the abdominal skin from the challenge sites was removed and fixed in 10% formalin solution. Briefly, the tissues were embedded in paraffin, cut into 5-µm-thick sections and then stained with either hematoxylin and eosin (H&E) to examine cell infiltration, or immunostained with rabbit anti-mouse CD4 (800-fold dilution; Bioss, MA, USA) or IL-31 (100-fold dilution; Abcam, Cambridge, UK) polyclonal antibodies for the analysis of T cells. Endogenous peroxidases were blocked using peroxide for 30 min at room temperature. To detect positive cells the sections were incubated with peroxidase-conjugated goat anti-rabbit IgG and visualized using diaminobenzidine solution using the Bond automatic system (Leica, Newcastle, UK). Inflammatory cell infiltrates were examined by light microscopy and corresponding images were captured using an Olympus BX51 microscopic/ DP71 Digital Camera System (Nagano, Japan).

## Measurement of *in vitro* cytokine production by For t 2-treated splenocytes

Splenocytes from the experimental mice were harvested on day 61 (48 h after the final challenge) and processed to form a single-cell suspension. Cells were cultured in 24-well flat-bottomed plates at a concentration of $1 \times 10^{6}$/ml and stimulated with 1 µg/ml *E*-rFor t 2 at 37˚C for 3–5 days. The culture supernatants were collected at each time interval and stored at -20˚C until required for the cytokine assay. The levels of IL-4, IL-10, IL-13 and interferon (IFN)-γ in the culture supernatants were measured using mouse ELISA kits (Life Technologies, Carlsbad, CA, USA) according to the manufacturer's instructions.

## RNA preparation and quantitative reverse transcription (qRT)-PCR of For t 2-treated splenocytes

Predesigned primer sequences [16] were used to measure the levels of cytokine mRNA expression. Briefly, on day 3 the total RNA from *E*-rFor t 2-stimulated splenocytes was purified using the Invitrogen TRIzol reagent (Life Technologies; Thermo Fisher Scientific, Inc.). cDNA was

obtained from 1 μg total RNA using a SuperScript III kit (Life Technologies). The reaction mixtures were amplified with SYBR Green PCR master mix (Life Technologies) in the presence of 0.2 μM of each of the specific primer sets. PCR amplification was conducted with an initial 10-min step at 95˚C followed by 40 cycles of 95˚C for 15 sec and 60˚C for 1 min using the StepOnePlus™ system (Applied Biosystems, CA, USA). The fluorescence signal was detected immediately after the extension step of each cycle, and the cycle at which the product was first detectable was recorded as the cycle threshold. Data were imported into an Excel database and analyzed using the comparative cycle threshold method with normalization of the raw data to β-actin.

## Statistical analysis

Data were presented as the mean ± standard error of the mean, and statistical analyses were performed using IBM SPSS Statistics software version 22 (IBM Corporation, Armonk, NY, USA) with appropriate methods. $P$-values $<0.05$ were considered to indicate a statistically significant difference.

## Results

### Mouse model and *in vitro* expression of the For t 2 DNA vaccine

Therapeutic schedule of For t 2 DNA vaccination on *E*-rFor t 2-sensitized mice is shown in Fig 1A. The endotoxin content of *E*-rFor t 2 after removing gel was under the detection limit (0.05 U/ml) of the *Limulus* Amebocyte Lysate test.

The full-length cDNA of For t 2 was cloned in the pCI-neo vector and the expression level of the construct was confirmed in HaCaT cells (a keratinocyte cell line from adult human skin). Culture supernatants from the transfected cells after 24 h were collected and analyzed via western blotting using a *E*-rFor t 2-specific antibody. The purity of *E*-rFor t 2 and the specificity of lab-made rabbit anti-*E*-rFor t 2 antibody were demonstrated in S1 Fig and described previously [3,15]. The expression of the 36 kDa For t 2 protein was detected and is shown in Fig 1B. No For t 2 protein was seen in the control sample transfected with the empty vector.

### Down-regulation of For t 2-specific IgE production and the ratio of IgG1/IgG2a after therapeutic For t 2 DNA vaccination

In the dose-response experiments, two consecutive shots of For t 2 DNA vaccine were given to *E*-rFor t 2-sensitized mice. ELISA was used to determine the effects of the DNA vaccine on the allergen specific-IgE, -IgG1 and -IgG2a levels in the sera on day 63. *E*-rFor t 2-induced specific IgE antibodies reduced by 40–56% in the V50, V100 and V200 groups compared with either the not-vaccinated group or the VO group (Fig 2A). However, only the change in the V50 group was statistically significant when compared with the VO group. For t 2-specific IgG2a, a surrogate maker of successful allergen-specific immunotherapy in mice, was elevated in all treatment groups in compared with not-vaccinated and VO groups, but no significant difference (Fig 2B). However, a significant decrease in the ratios of IgG1/IgG2a was observed in all For t 2 DNA-vaccinated groups compared with the VO control group (Fig 2C).

### Therapeutic For t 2 DNA vaccine ameliorates scratching bouts induced by midge allergen challenge

Post-sensitization vaccination was administered to all groups of mice and comprised 3 ID challenges with *E*-rFor t 2 protein during days 56–61. After the final challenge, the scratching behaviors (the surrogate marker of an itch) were immediately videotaped for 1 h. In the not-

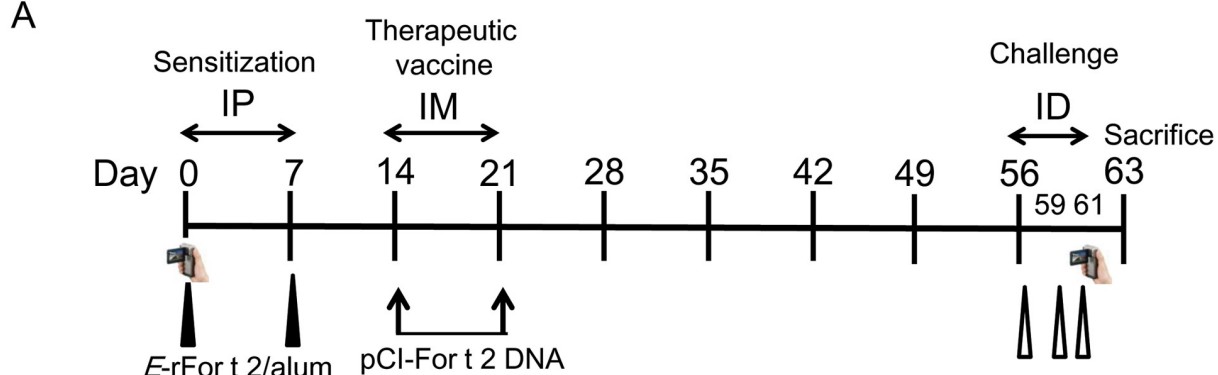

A

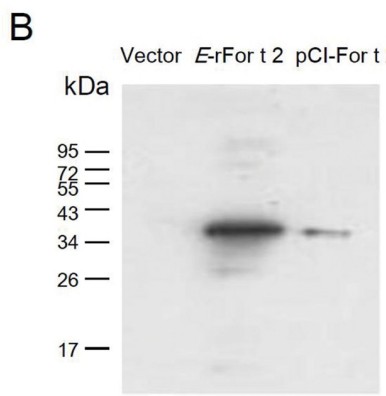

| Groups | No. | Sensitization (IP) | Therapeutic regimen (IM) | Challenge (ID) |
|---|---|---|---|---|
| Sentinel | 4 | PBS | PBS | PBS |
| Nonvaccinated | 6 | *E*-rFor t 2 | PBS | *E*-rFor t 2 |
| VO (Vector only) | 6 | *E*-rFor t 2 | 100 µg pCI-neo plasmid | *E*-rFor t 2 |
| V50 | 6 | *E*-rFor t 2 | 50 µg For t 2 DNA vaccine | *E*-rFor t 2 |
| V100 | 6 | *E*-rFor t 2 | 100 µg For t 2 DNA vaccine | *E*-rFor t 2 |
| V200 | 6 | *E*-rFor t 2 | 200 µg For t 2 DNA vaccine | *E*-rFor t 2 |

**Fig 1. Experimental procedure.** (A) Therapeutic regimen of For t 2 DNA vaccination in *E*-rFor t 2-sensitized mice. BALB/c mice received 2 intra-peritoneal injections of *E*-rFor t 2/alum on days 0 and 7. Subsequently, the sensitized mice were treated intra-muscularly with For t 2 DNA vaccine on days 14 and 21. From days 56 to 61 the mice were challenged intra-dermally (ID) with 3 doses of *E*-rFor t 2 on 3 consecutive days. Blood samples were collected bi-weekly from the retro-orbital venous plexus. The scratching behaviors of the mice were videotaped at days 0 and 61. All experimental mice were sacrificed on day 63. (B) Western blot analysis of culture supernatant from HaCaT cells transfected with empty vector or pCI-For t 2 using rabbit anti-*E*-rFor t 2 antibodies. *E. coli*-expressed recombinant protein was seen at 36 kDa as detected in the positive control.

vaccinated group and the VO group, the mean number of scratching bouts was >300 in the initial 20 min (Fig 3). All For t 2 DNA vaccine-treated groups showed significantly fewer scratching bouts on the challenge sites during the first 20 min compared with both the not-vaccinated group and the VO group. Treatment of For t 2-sensitized mice with the For t 2 DNA vaccine significantly inhibited the incidence of skin itching, which is the most annoying problem for human allergy patients.

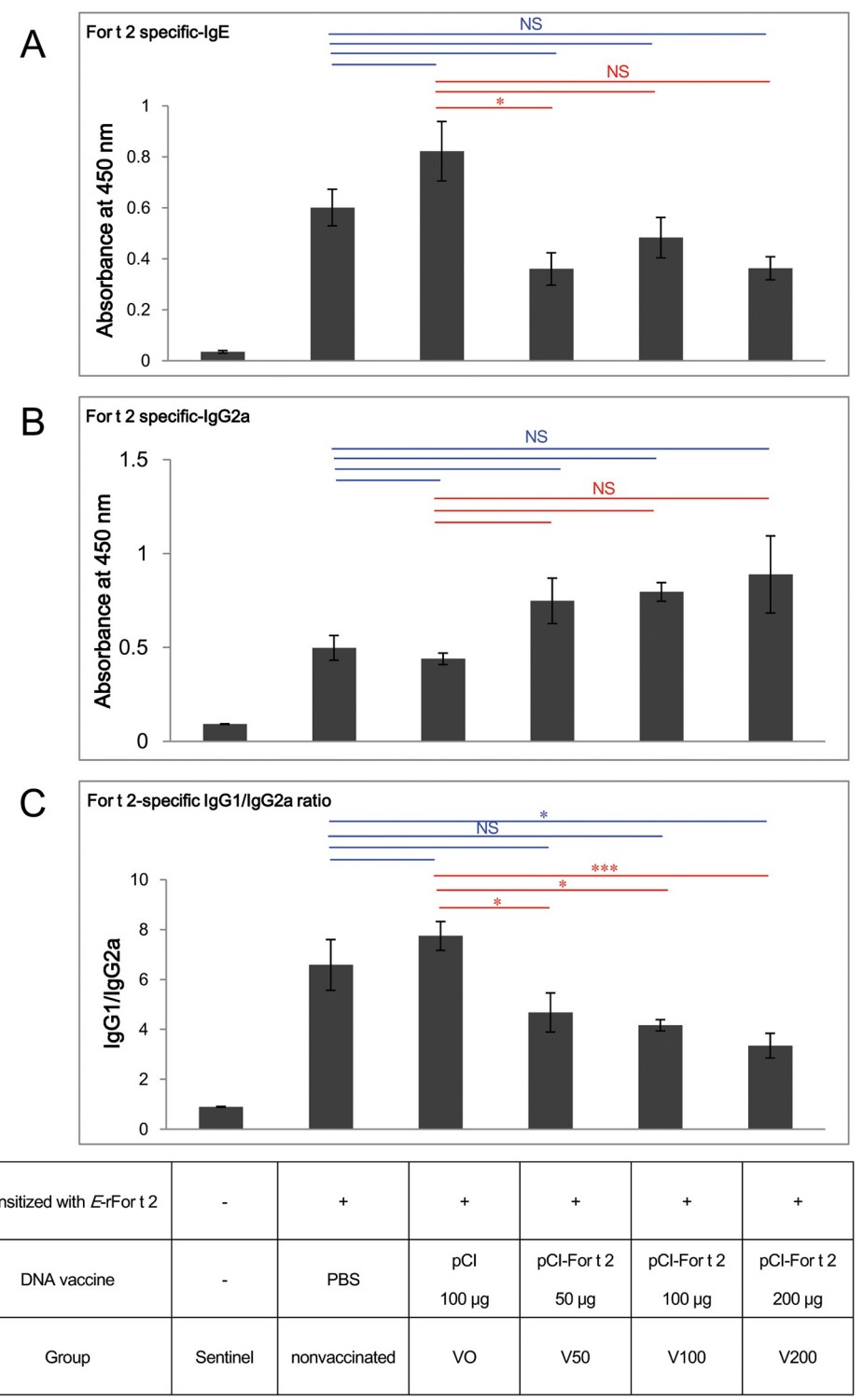

| Sensitized with *E*-rFor t 2 | - | + | + | + | + | + |
|---|---|---|---|---|---|---|
| DNA vaccine | - | PBS | pCI 100 µg | pCI-For t 2 50 µg | pCI-For t 2 100 µg | pCI-For t 2 200 µg |
| Group | Sentinel | nonvaccinated | VO | V50 | V100 | V200 |

**Fig 2. Serum levels of For t 2-specific antibodies on week 9 as determined by ELISA.** (A) IgE, (B) IgG2a and (C) the ratio of IgG1/IgG2a. The statistics show comparisons between the nonvaccinated or vector only groups and the DNA vaccine groups by one-way analysis of variance with Bonferroni multiple range test. *$p < 0.05$, ***$p < 0.001$; NS, not statistically significant.

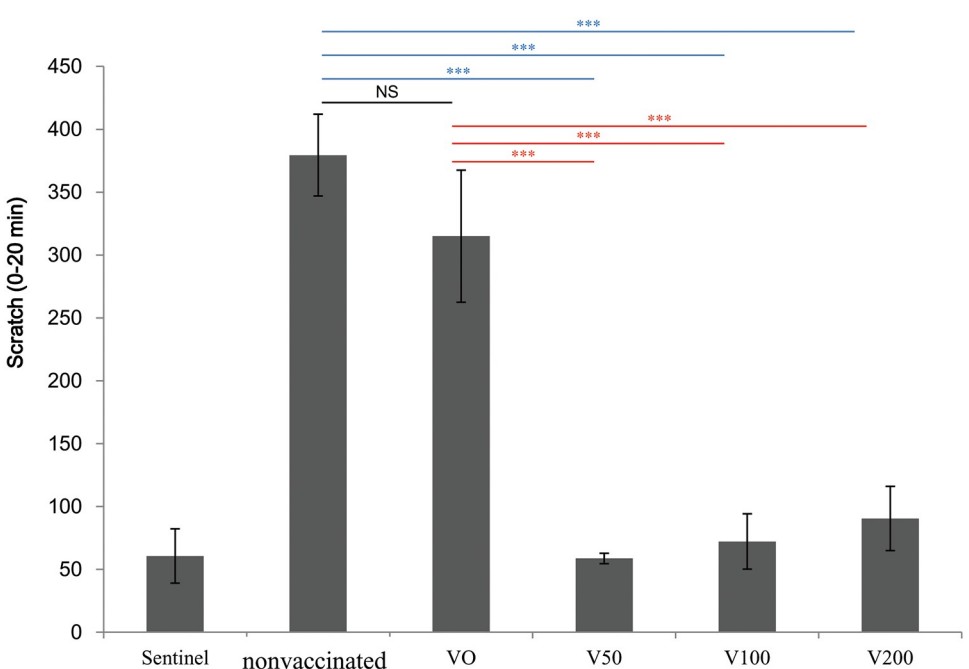

**Fig 3. The number of scratching bouts induced in mice following administration of *E*-rFor t 2 by intradermal injection.** The statistics show comparisons between the not-vaccinated and VO groups with the DNA vaccine groups using Dunnett's *t*-test. ***$p < 0.001$; NS, not statistically significant.

## Effect of the For t 2 therapeutic DNA vaccine on cytokine gene expression and the production of rFor t 2-stimulated splenocytes

The spleen plays an important role in humoral and cellular immune responses. It contains a variety of immune cells which provide the organ with a cytokine-rich environment [17]. Splenocytes were harvested from all groups of mice and stimulated with rFor t 2 allergen to determine whether Th1 or Th2 cytokines were affected by the For t 2 DNA vaccination. The cytokine mRNA expression on day 3 and the protein levels on day 5 in response to rFor t 2 were determined using qRT-PCR and ELISA, respectively. Significant down regulation of IL-13 mRNA expression and protein secretion was only detected only in the V50 group compared with the not-vaccinated group (Fig 4A). There was a marked decrease in IL-4 mRNA expression in the V50 and V100 groups compared with the nonvaccinated group. The protein levels of IL-4 were >500 pg/ml in the not-vaccinated group. All DNA vaccine-treated groups produced extremely low amounts of IL-4 protein, which were below the detection limits (Fig 4B). The mRNA level of IL-10 was only significantly increased in the V50 group compared with the nonvaccinated group, and there were no significant differences in IL-10 protein secretion between all groups (Fig 4C). The levels of IFN-γ mRNA and protein expression showed no significant changes between all groups (Fig 4D).

## Effect of the For t 2 therapeutic DNA vaccine on delayed skin reactions

Skin tissue from the challenged sites was harvested 48 h after the final ID challenge with the *E*-rFor t 2 allergen, and paraffin sections were prepared for histopathology. Fig 5A and 5B shows that the total number of infiltrate cells, mainly the eosinophils, decreased significantly in the V50, V100 and V200 groups compared with the nonvaccinated and VO groups. IL-31 is known to be one of the key cytokines that induces itching and promotes scratching in mouse

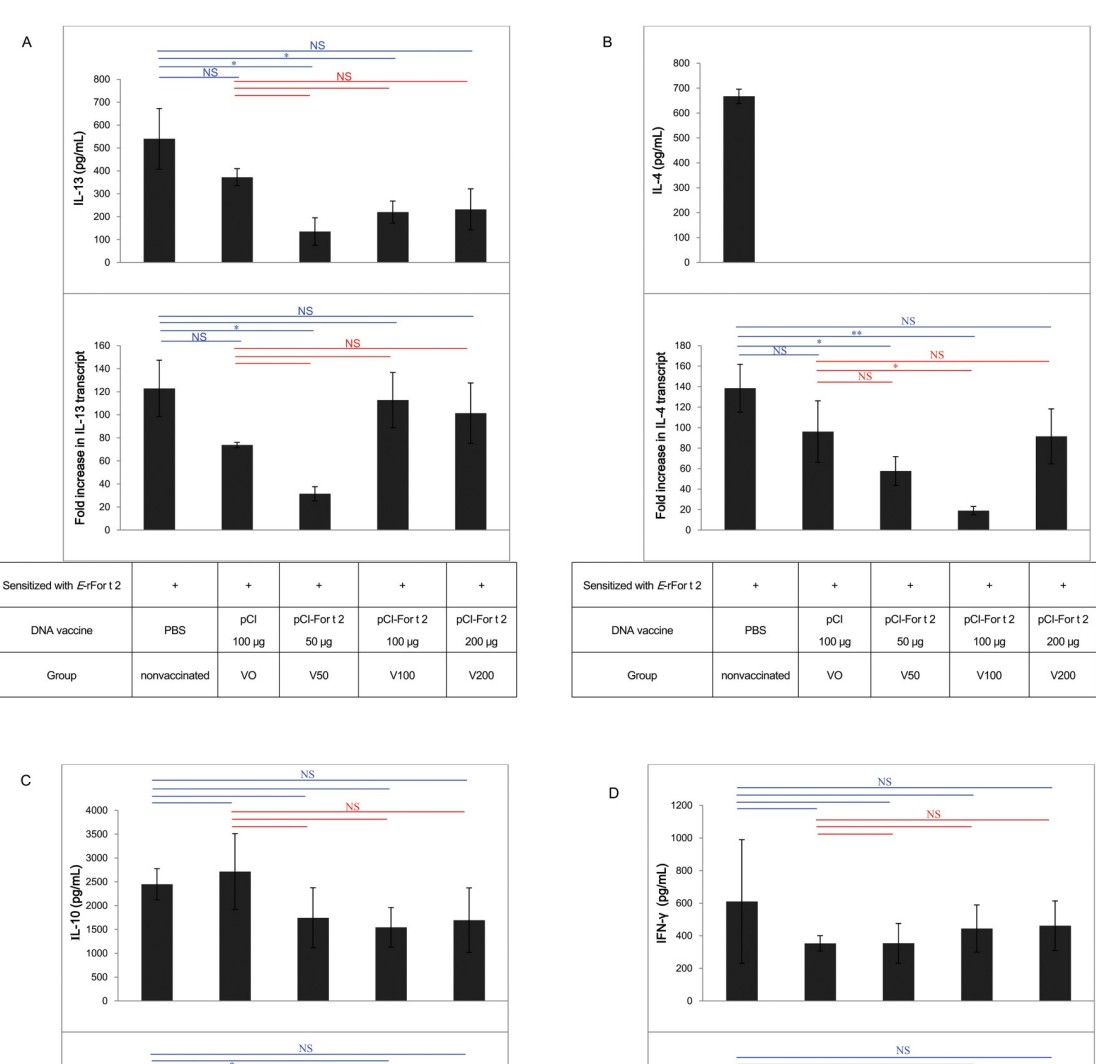

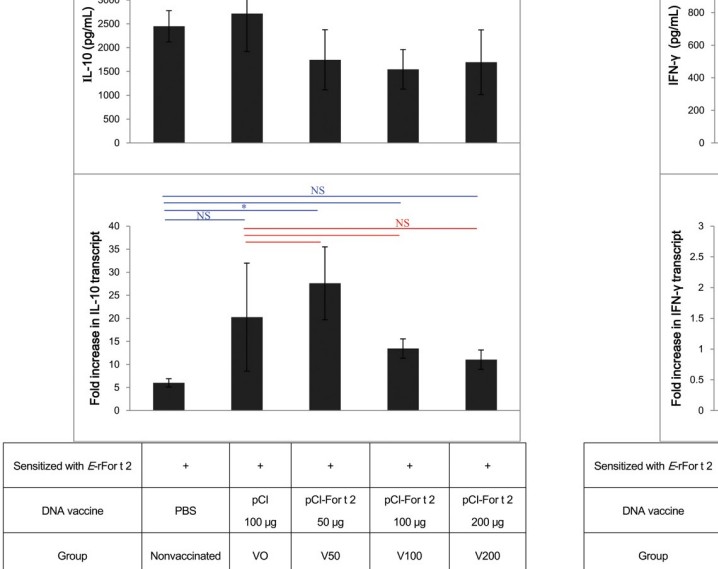

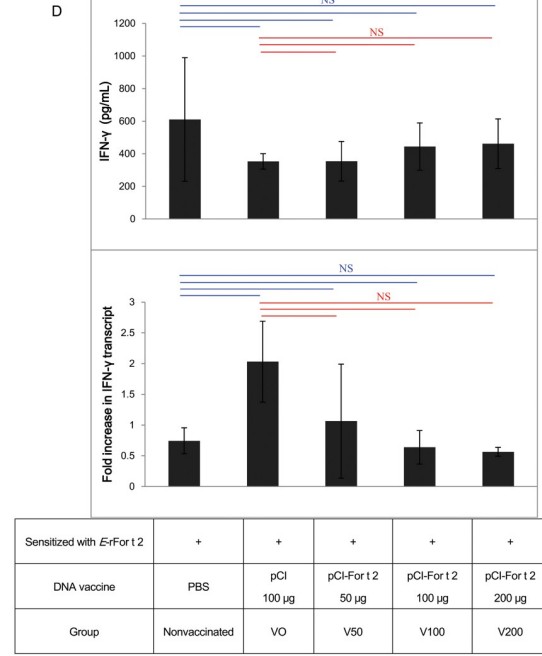

**Fig 4. Cytokine protein expression and the mRNA levels of splenocytes from each group of mice as determined by ELISA and qRT-PCR, respectively.** (A) IL-13, (B) IL-4, (C) IL-10 and (D) IFN-γ. Cytokine release data are shown as the mean ± SEM from 3 independent experiments. qRT-PCR data are expressed as the mean fold increase ± SEM from 3 independent experiments. The statistical significance of differences among the groups was assessed by one-way analysis of variance with Dunnett's *t*-test. $^*p<0.05$; $^{**}p<0.01$; NS, not statistically significant; SEM, standard error of the mean.

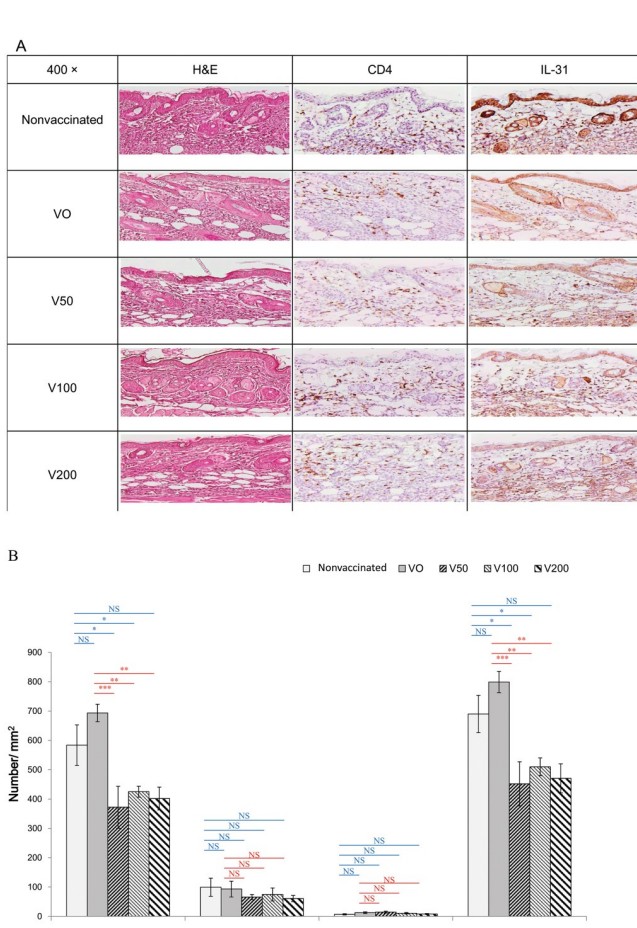

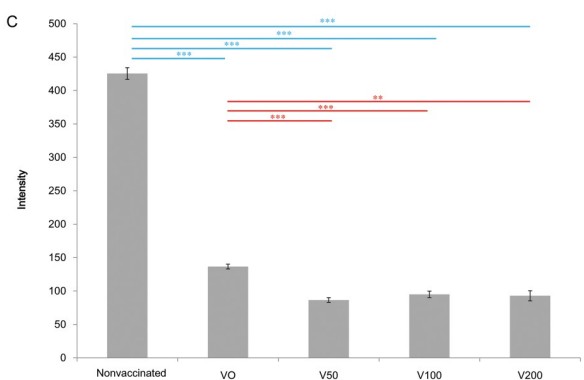

**Fig 5. Skin histopathology of mice from different groups after challenge with rFor t 2 allergen.** (A) Representative sections of the abdominal skin was obtained 48 h after the intradermal challenge test and stained with H&E or with rabbit anti-mouse CD4 and IL-31 antibodies. The sections were observed under a 400x light microscope. (B) The infiltrating inflammatory cells were quantified in the H&E-stained skin. The results are expressed as the mean ± SEM of the cell number per square mm of skin specimen. (C) The intensity levels of IL-31 in the epidermis from (A) as determined by Gel-Pro image software. The statistical significance of differences among the groups was assessed by one-way analysis of variance with Dunnett's $t$-test. $^*p<0.05$, $^{**}p<0.01$, $^{***}p<0.001$; NS, not statistically significant.

models of atopic dermatitis [18]. As shown at Fig 5C, expression of intradermal IL-31 was significantly decreased in all of the DNA vaccinated groups, including the VO group, compared with the nonvaccinated group. However, the reduction of IL-31 is more significant in the V50, V100 and V200 groups compared with the VO group.

### Therapeutic effects of the For t 2 DNA vaccine in crude midge extract-sensitized mice

In a second set of experiments (n = 6 per group), the mice were sensitized with crude midge extracts (containing all midge allergens) and then treated with the For t 2 DNA vaccine (Fig 6A). Administration of the For t 2 DNA vaccine at 100 or 200 μg ameliorated the midge allergen-induced scratch bouts (Fig 6B), as well as midge allergen-induced IL-13 production from splenocytes (Fig 6C), and inflammatory cell infiltration in the lesions 48 h after ID challenge.

A

| | Group | Sensitization (IP) | Therapy (IM) | Challenge (ID) |
|---|---|---|---|---|
| 1 | Sentinel | PBS | PBS | PBS |
| 2 | Nonvaccinated | Crude midge extract | PBS | Crude midge extract |
| 3 | VO | Crude midge extract | 100 μg Vector only | Crude midge extract |
| 4 | V100 | Crude midge extract | 100 μg For t 2 vaccine | Crude midge extract |
| 5 | V200 | Crude midge extract | 200 μg For t 2 vaccine | Crude midge extract |

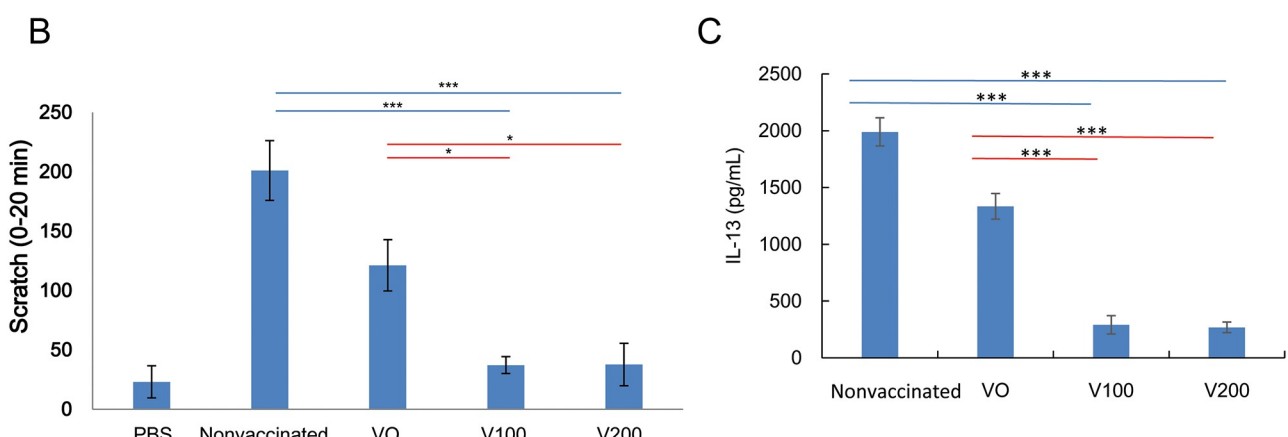

**Fig 6. Therapeutic effects of For t 2 DNA vaccines on crude midge extract-sensitized mice.** (A) Experimental groups of therapeutic For t 2 DNA vaccinations on crude midge extract-sensitized mice. (B) Scratching bouts induced by intradermal challenge of crude midge extract. (C) IL-13 protein levels of midge extract stimulated-splenocytes from each group of mice, as determined by ELISA. The statistical significance of differences among the groups was assessed by one-way analysis of variance with Dunnett's $t$-test. $^{*}p{<}0.05$, $^{***}p{<}0.001$.

### Effects of the booster doses and CpG adjuvant (TLR-9 ligand) on the For t 2 therapeutic DNA vaccine

An additional experiment was designed to investigate whether additional booster doses (increasing the number of injections from 2 doses to 4 doses), or adding CpG adjuvant (TLR-9 ligand) had beneficial therapeutic effects on the For t 2 DNA vaccine. It was found that after vaccination, decreasing levels of For t 2-specific IgE and increasing levels of specific IgG2a were only observed in group V50-2 (2 doses) but not the CPG groups or the V50-4 groups (four doses). The number of scratching bouts in mice induced by ID challenge of rFor t 2 allergen only decreased in the V50-2 group but not in the CGP groups or the V50-4 groups. These results suggest that 2 additional doses and the CpG adjuvant do not provide additional benefits to 2 doses of the For t 2 therapeutic DNA vaccine. Taken together, these findings suggest that two shots of 50–100 μg DNA is sufficient to improve biting midge-induced allergic symptoms and inflammation.

## Discussion

In the present study, the therapeutic effect of the For t 2 DNA vaccine was investigated in mice with an established biting midge allergy. It was found that two doses of For t 2 DNA vaccine against the major allergen of biting midge *F. taiwana*, attenuated itching and allergic inflammation in mice with an established biting midge allergy. After treatment with two doses of For t 2 DNA vaccine, the mice that were already sensitized to biting midges demonstrated a significant decrease in the number of scratching bouts as well as the secretion of type II cytokines, such as IL-13 and IL-4; skin eosinophil infiltration and intra-dermal IL-31 production was also significantly reduced in response to ID midge-allergen challenge. To the best of our knowledge, this is the first report on a therapeutic DNA vaccine treating established biting insect allergies.

IL-31 is a cytokine principally produced by activated Th2 cells. IL-31 receptors are constitutively expressed on epithelial cells and keratinocytes [18–20]. There is mounting evidence showing that IL-31 is correlated with the itchy pathophysiology of atopic dermatitis, cutaneous T-cell lymphoma, uremic pruritus, allergic contact dermatitis and chronic urticaria [21,22]. In the current study, For t 2-specific IgE, the For t 2 specific IgG1/IgG2a ratio, IL-4, IL-13 and skin inflammation were all reduced following administration of the DNA vaccine, there was also an increase in For t 2-specific IgG2a, as one would expect to see in a successful allergen specific immunotherapy [23,24]. In addition to this, allergen-induced intra-dermal IL-31 production was significantly reduced following the administration of DNA vaccine therapy. This change in IL-31 supported the significant reduction in the number of itchy scratching bouts observed in the mice after therapy in the present study.

Interestingly, the optimal number of doses for this therapeutic For t 2 DNA vaccine was two consecutive doses one week apart, which is the same as previously recommended for the prevention of the development of biting midge allergies [15]. Increasing the number of injections to four doses or adding CPG adjuvants seemed to have no benefits. Our data also showed that increasing the dose of vaccine from 50 to 100 or 200 μg did not seem to influence the therapeutic effects. Fifty μg was capable of achieving the same clinical effects as those achieved by 100 and 200 μg. As our previous study [15] showed that 25 μg had less effect than 50 μg in preventing biting midge allergy, it does not seem to be the case that any dose is effective. A possible explanation may be the capacity limitation of cells at injection sites to express the target protein.

Although there are at least 11 allergens in the biting midge *F. taiwana* [2], a DNA vaccine encoding the single major allergen For t 2, seems to be adequate to alleviate the itchy scratching bouts, as well as allergen-induced IL-13 production in mice sensitized with whole crude midge allergens. These results imply that it may not be necessary to use multiple DNA

sequences of midge allergens in the future when applying this DNA vaccine to human midge allergy subjects.

There are limitations to the current study. First, whether the therapeutic effects of the DNA vaccine as observed in mice could be generalized to humans remains unknown. It is known that DNA-based vaccines tend to have weaker immunogenicity in primates and humans compared with rodents, particularly when administered by conventional injection. Further studies on primates may be necessary prior to human trials. Second, conventional intra-muscular injection was used in the current study. Using alternative routes of administration, such as electroporation, have been reported to enhance the effect of a DNA vaccine encoding the major house dust allergen Der p 2 [25]. Third is the issue of adjuvants. Using the CpG motif as an adjuvant in the present study provided no additional benefits to the vaccine. However, other adjuvants were not tested. For example poly-L-lysine (PLL), which has been reported to improve the prophylactic and therapeutic effect of the peanut allergen Ara h 2 DNA vaccine in peanut-allergic mice [26]. Furthermore, whether the For t 2 DNA vaccine can last long enough to cover the whole midge season requires further investigation.

In conclusion, two doses of allergen specific-immunotherapy administered one week apart, using DNA encoding the major midge allergen For t 2, is effective at treating mice with established biting midge allergies. The treatment effects are evident by clinical, immunological and histopathological improvements. With its convenience and low cost the For t 2 DNA vaccine could be a favorable strategy for the treatment of this problem in human patients.

## Supporting information

**S1 Fig. The purity of *E*-rFor t 2 and the specificity of lab-made rabbit IgG against recombinant and natural For t 2 proteins.** (A) Coomassie Blue-stained SDS-PAGE of purified *E.coli*-expressed For t 2 recombinant protein. (B) Immunoblotting of *E*-rFor t 2 or (C) midge extracts with rabbit anti-*E*-rFor t 2 polyclonal antibodies (lane 3). Lane 1, midge extract probed with non-immunized rabbit serum; lane 2, midge extract probed with pre-immunized rabbit serum.
(TIF)

**S1 Raw images.**
(PDF)

## Author Contributions

**Conceptualization:** Yi-Hsing Chen.

**Data curation:** Mey Fann Lee, Yi-Hsing Chen, Pei-Pong Song, Tzu-Mei Lin.

**Formal analysis:** Mey Fann Lee, Tzu-Mei Lin.

**Funding acquisition:** Yi-Hsing Chen.

**Investigation:** Mey Fann Lee, Pei-Pong Song, Tzu-Mei Lin.

**Methodology:** Mey Fann Lee, Pei-Pong Song, Tzu-Mei Lin.

**Project administration:** Mey Fann Lee, Yi-Hsing Chen, Tzu-Mei Lin.

**Supervision:** Yi-Hsing Chen.

**Validation:** Yi-Hsing Chen, Pei-Pong Song.

**Visualization:** Yi-Hsing Chen, Pei-Pong Song.

**Writing – original draft:** Mey Fann Lee, Yi-Hsing Chen.

**Writing – review & editing:** Yi-Hsing Chen.

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
