## [Decision Letter · Decision Letter 0]

21 Feb 2020

PONE-D-20-00004

Therapeutic DNA vaccine attenuates itching and allergic inflammation in mice with established biting midge allergy

PLOS ONE

Dear Prof. Chen,

Thank you for submitting your manuscript to PLOS ONE. After careful consideration, we feel that it has merit but does not fully meet PLOS ONE’s publication criteria as it currently stands. Therefore, we invite you to submit a revised version of the manuscript that addresses the points raised during the review process.

We would appreciate receiving your revised manuscript by Apr 06 2020 11:59PM. To enhance the reproducibility of your results, we recommend that if applicable you deposit your laboratory protocols in protocols.io, where a protocol can be assigned its own identifier (DOI) such that it can be cited independently in the future. For instructions see: http://journals.plos.org/plosone/s/submission-guidelines#loc-laboratory-protocols

We look forward to receiving your revised manuscript.

Kind regards,

Junji Xing, Ph.D.

Academic Editor

PLOS ONE

Journal Requirements:

2. To comply with PLOS ONE submission requirements, in your Methods section, please provide the precise method of sacrifice and ensure that you have reported any other efforts undertaken to alleviate suffering.

Reviewers' comments:

Reviewer's Responses to Questions

**Comments to the Author**

1. Is the manuscript technically sound, and do the data support the conclusions?

Reviewer #1: Yes

Reviewer #2: No

2. Has the statistical analysis been performed appropriately and rigorously? 

Reviewer #1: Yes

Reviewer #2: No

3. Have the authors made all data underlying the findings in their manuscript fully available?

Reviewer #1: Yes

Reviewer #2: No

4. Is the manuscript presented in an intelligible fashion and written in standard English?

Reviewer #1: No

Reviewer #2: Yes

5. Review Comments to the Author

Reviewer #1: This manuscript by Chen et al examines the potential therapeutic benefit of a DNA vaccine against the biting midge-Forcipomyia taiwana. This is of importance regionally as this is the most prevalent biting insect allergy in Taiwan, with t2 being the major allergen component. Using a mouse model, they aimed to show if a DNA vaccine agains For t2 allergens could prevent or attenuate various readouts for the allergen response, namely, T2 cytokines and the physical readout of scratching. Overall, they discover there appears to be some benefit to this vaccine when delivered in the proper dosage to mice. This has implications that further studies could be useful for humans as treatment-also noted that this would be reasonably affordable option for treatment as well. These studies, while of importance regionally, would be less relevant to those outside of Taiwan.

The abstract needs editing for better grammar.

Figure legends are scattered throughout the methods and results sections making it awkward to read. These should be placed in the appropriate locations as directed by the journal.

A discussion of potential reasons contributing to why increasing dosing of vaccine had less activity against the midge allergy should be discussed.

When analyzing scratching behavior from video tapes, were the persons assessing blinded to the groups?

Reviewer #2: Lee et al., present a manuscript seeking to assess the impact of therapeutic vaccination with a midge protein (For t 2) to protect from allergic reaction in mice. The data, as presented, are inconclusive and somewhat confounding. Some of the terminology in the text needs to be corrected for accuracy. A number of methods are incomplete, antigens employed not completely described and key antisera is not sufficiently defined. Resolution of major concerns will require additional experiments focused on demonstrating that For t 2 in crude midge extracts and in recombinant form are relevant target of vaccination.

Materials and Methods section:

“sterilized with a 0.22 um syringe filter” is not the correct terminology when material is passed through a 0.22 um filter owing to the filterability of most viruses through this pore size. “passed through” would be appropriate.

The detection limit of the endotoxin test should be stated where the results of the test are described, which should be at the beginning of the Results section. Currently, no description is included.

The description of Figure 1 should be moved to the Results section. The time post transfection when cells were collected and lysed for panel B should be more completely described. Also, E-rFor t 2 should be defined as the “E. coli-expressed recombinant protein”. In addition to the immunoblot, a gel stained for total protein should be shown to demonstrate the purity of the E-rFor t 2. Given that “rabbit anti-rFor t 2 polyclonal antibodies” are employed to show that the 36 kDa protein in the gel is antigenically related to For t 2, the source and reference for the antibody must be better documented. Evidence that the material produced in bacteria or following DNA transfection of HaCaT cells are related to natural antigen is crucial here, particularly if the polyclonal sera used was raised by immunizing with these same recombinant sources of protein.

Results section:

Figure 1 should start this section. Where “rFor t 2” is written, “E-rFor t 2” should be used to be consistent, Both Figure 1A and Figure 2 currently indicate “rFor t 2”. The term “not-vaccinated” should be corrected to “nonvaccinated”.

If the source of “rFor t 2 used in ELISA assays is the same as used for sensitization of the animals, more need to be included to demonstrate that the bacterially produced antigen is pure.

It would be appropriate to use a different source of recombinant protein for the ELISA to avoid cross reactivity due to undisclosed impurities.

Figure 3 attempts to show that post-sensitization vaccination can reduce inching, but the description “Five weeks after For t 2 DNA vaccination, all groups of mice received 3 ID challenges with rFor t 2 protein during days 56-61” does not state that animals were first sensitized by IP inoculation as shown in Figure 1A. This must be the case for the controls to have shown high levels of scratching, but the text description must be unambiguous. This data suggests that any dose of therapeutic vaccine (V50, V100 or V200) tested was sufficient to reduce scratching to a similar degree.

Authors do not provide a rationale or evidence that the spleen is the appropriate organ to assess for cytokines in this model. Results observed are confounding, likely due to an incomplete experimental strategy.

Figure 4 shows that, in splenocytes, cytokines IL-13, IL-4 and IL-10 showed consistent patterns by ELISA at any dose of vaccine tested, although there is a surprising absence of IL-4 in a control group as well, making this set of observations impossible to interpret. IFNg or IFN mRNA did not show any differences between groups. IL-13 mRNA was only significantly reduced in the V50 group; whereas IL-4 mRNA was reduced most in the V100 group and IL-10 mRNA was only elevated in the V50 group. The number of times this experiment was performed is not stated, but none of these peculiarities seems likely. The patterns observed seem arbitrary.

Figure 5 shows that eosinophils are higher in controls than any of the For t 2 vaccinated groups. in situ immunostaining in skin sections for CD4+ cells and intradermal IL-31. Patterns are referred to a “significantly” changed, without any possibility of statistical significance in this type of experiment. Furthermore, intensity of IL-31 staining is quantified in Fig 5C, with statistics applied inappropriately, but showing high significance. Text and statistics must be better aligned with the type of experiment shown.

Figure 6 shows an experiment where groups of mice are sensitized with crude midge extract and then vaccinated before challenge with crude midge extract. Vaccination is shown to reduce scratching and IL-13 levels.

In order to have a more rigorous series of animal experiments, two additional experiments are needed, where E-rFor t 2 is used to sensitize and crude midge extract is used to challenge, either with or without vaccination, and also where crude midge extract is used to sensitize and E-For t 2 is used to challenge. These variations will help address some of the concerns about antigens that have been applied here.

6. PLOS authors have the option to publish the peer review history of their article (what does this mean?). If published, this will include your full peer review and any attached files.

Reviewer #1: No

Reviewer #2: No

---

## [Author Response · Author response to Decision Letter 0]

22 Mar 2020

Reviewer #1: This manuscript by Chen et al examines the potential therapeutic benefit of a DNA vaccine against the biting midge-Forcipomyia taiwana. This is of importance regionally as this is the most prevalent biting insect allergy in Taiwan, with t2 being the major allergen component. Using a mouse model, they aimed to show if a DNA vaccine against For t2 allergens could prevent or attenuate various readouts for the allergen response, namely, T2 cytokines and the physical readout of scratching. Overall, they discover there appears to be some benefit to this vaccine when delivered in the proper dosage to mice. This has implications that further studies could be useful for humans as treatment-also noted that this would be reasonably affordable option for treatment as well. These studies, while of importance regionally, would be less relevant to those outside of Taiwan.

Question 1. The abstract needs editing for better grammar.

Reply: The abstract has been sent for English editing by a professional English editor as per the reviewer’s suggestion.

Question 2. Figure legends are scattered throughout the methods and results sections making it awkward to read. These should be placed in the appropriate locations as directed by the journal.

Reply: All figure legends have been moved to the last part of the manuscript, from page 23 to 26. 

Question 3. A discussion of potential reasons contributing to why increasing dosing of vaccine had less activity against the midge allergy should be discussed.

Reply: We have added the following to the discussion on page 16 “…. Our data also showed that increasing the dose of vaccine from 50 to 100 or 200 �g did not seem to influence the therapeutic effects. Fifty �g was capable of achieving the same clinical effects as those achieved by 100 and 200 �g. As our previous study [15] showed that 25��g had less effect than 50��g in preventing biting midge allergy, it does not seem to be the case that any dose is effective. A possible explanation may be the capacity limitation of cells at the injection site to express the target protein.” 

Question 4. When analyzing scratching behavior from video tapes, were the persons assessing blinded to the groups?

Reply: Yes. This has been added to the revised manuscript. Please see page 7, line 16. “The observation of scratching behavior was blinding measured and performed as previously described [15,16].”

Reviewer #2: Lee et al., present a manuscript seeking to assess the impact of therapeutic vaccination with a midge protein (For t 2) to protect from allergic reaction in mice. The data, as presented, are inconclusive and somewhat confounding. Some of the terminology in the text needs to be corrected for accuracy. A number of methods are incomplete, antigens employed not completely described and key antisera is not sufficiently defined. Resolution of major concerns will require additional experiments focused on demonstrating that For t 2 in crude midge extracts and in recombinant form are relevant target of vaccination.

Materials and Methods section:

Question 1. “sterilized with a 0.22 um syringe filter” is not the correct terminology when material is passed through a 0.22 um filter owing to the filterability of most viruses through this pore size. “passed through” would be appropriate.

Reply: The term on page 5 line 8 has been changed to “passed through” as per the suggestion of the reviewer.

Question 2. The detection limit of the endotoxin test should be stated where the results of the test are described, which should be at the beginning of the Results section. Currently, no description is included.

Reply: A description of the detection limit of the endotoxin has been added to Page 5, Lines 9-11: “The endotoxin content was determined using an E-TOXATE kit (Sigma-Aldrich; Merck KGaA, Darmstadt, Germany). The lowest detection limit of the test was 0.05 endotoxin units per ml.” The following was also added to Page 10, Lines 10-11: “The endotoxin content of E-rFor t 2 after removing gel was under the detection limit (0.05 U/ml) of the Limulus Amebocyte Lysate test.”

Question 3. 1) The description of Figure 1 should be moved to the Results section. The time post transfection when cells were collected and lysed for panel B should be more completely described. 2) Also, E-rFor t 2 should be defined as the “E. coli-expressed recombinant protein”. 3) In addition to the immunoblot, a gel stained for total protein should be shown to demonstrate the purity of the E-rFor t 2. Given that “rabbit anti-rFor t 2 polyclonal antibodies” are employed to show that the 36 kDa protein in the gel is antigenically related to For t 2, the source and reference for the antibody must be better documented. Evidence that the material produced in bacteria or following DNA transfection of HaCaT cells are related to natural antigen is crucial here, particularly if the polyclonal sera used was raised by immunizing with these same recombinant sources of protein.

Reply: 1) The description of Figure 1 has been moved to the first part of the Results section, Page 10, Lines 9-11. 2) All descriptions of E-coli-expressed For t 2 protein in this manuscript have been changed to “E-rFor t 2”. 3) A supplemental figure, Fig S1, has been added to demonstrate the purity of E-rFor t 2 (Lane A and B) as well as the specificity of rabbit anti-rFor t 2 polyclonal antibodies to 36 kDa natural For t 2 protein from whole midge extract (Lane C).

Fig S1. The purity of E-rFor t 2 and the specificity of lab-made rabbit IgG against recombinant and natural For t 2 proteins. (A) Coomassie Blue-stained SDS-PAGE of purified E.coli-expressed For t 2 recombinant protein. (B) Immunoblotting of E-rFor t 2, or (C) midge extracts with rabbit anti-E-rFor t 2 polyclonal antibodies (lane 3). Lane 1, midge extract probed with non-immunized rabbit serum; lane 2, midge extract probed with pre-immunized rabbit serum. 

Results section:

Question 4. 1) Figure 1 should start this section. 2) Where “rFor t 2” is written, “E-rFor t 2” should be used to be consistent, Both Figure 1A and Figure 2 currently indicate “rFor t 2”. 3) The term “not-vaccinated” should be corrected to “nonvaccinated”.

Reply: 1) The description of Figure 1 has been moved to the first part of the Results section, Page 10 Lines 9-10. 2) All descriptions of E-coli-expressed For t 2 protein in this manuscript have been changed to “E-rFor t 2”. 3) The term “not-vaccinated” has been changed to “nonvaccinated”, as shown in figure 1A, and in all related parts of the manuscript.

Question 5. If the source of “rFor t 2 used in ELISA assays is the same as used for sensitization of the animals, more need to be included to demonstrate that the bacterially produced antigen is pure. It would be appropriate to use a different source of recombinant protein for the ELISA to avoid cross reactivity due to undisclosed impurities.

Reply: A supplemental figure, Fig S1, has been added to demonstrate the purity of E-rFor t 2 (panel A and B).

Question 6. Figure 3 attempts to show that post-sensitization vaccination can reduce inching, but the description “Five weeks after For t 2 DNA vaccination, all groups of mice received 3 ID challenges with rFor t 2 protein during days 56-61” does not state that animals were first sensitized by IP inoculation as shown in Figure 1A. This must be the case for the controls to have shown high levels of scratching, but the text description must be unambiguous. This data suggests that any dose of therapeutic vaccine (V50, V100 or V200) tested was sufficient to reduce scratching to a similar degree.

Reply: We apology for the confusion. For greater clarity, the description has been rephrased as follows: “Post-sensitization vaccination was administered to all groups of mice and comprised 3 ID challenges with E-rFor t 2 protein during days 56-61.” Please see Page 11, Lines 17-18. 

As our previous study (MF Lee, et al. Allergy 71 (2016) 522–531) showed that 25��g had less effect than 50��g in preventing biting midge allergy, it does not seem to be the case that any dose is effective. The following sentences have been added to the Discussion section, Page 16 and 17: “Our data also showed that increasing the dose of vaccine from 50 to 100 or 200 �g did not seem to influence the therapeutic effects. Fifty �g was capable of achieving the same clinical effects as those achieved by 100 and 200 �g. As our previous study [15] showed that 25��g had less effect than 50��g in preventing biting midge allergy, it does not seem to be the case that any dose is effective. A possible explanation may be the capacity limitation of cells at injection sites to express the target protein.” 

Question 7. Authors do not provide a rationale or evidence that the spleen is the appropriate organ to assess for cytokines in this model. Results observed are confounding, likely due to an incomplete experimental strategy. 

Reply: In order to obtain sufficient cells for experiments, especially for those requiring further culture and stimulation with allergen, we used splenocytes from mice as a surrogate for peripheral mononuclear cells in humans. We have add a description and a reference (Mebius RE, Kraal G. Nat Rev Immunol. 2005; 5: 606-616. ) on Page 12 to explain the rationale for using splenocytes, as follows: “The spleen plays an important role in humoral and cellular immune responses. It contains a variety of immune cells, which provide the organ with a cytokine-rich environment.”.

Question 8. Figure 4 shows that, in splenocytes, cytokines IL-13, IL-4 and IL-10 showed consistent patterns by ELISA at any dose of vaccine tested, although there is a surprising absence of IL-4 in a control group as well, making this set of observations impossible to interpret. IFNg or IFN mRNA did not show any differences between groups. IL-13 mRNA was only significantly reduced in the V50 group; whereas IL-4 mRNA was reduced most in the V100 group and IL-10 mRNA was only elevated in the V50 group. The number of times this experiment was performed is not stated, but none of these peculiarities seems likely. The patterns observed seem arbitrary. 

Reply: We believe it is reasonable to suppose that a reduction in scratching bouts after challenge (equivalent to a reduction in itching sensation in humans after midge bites) would indicate that the DNA vaccine had a demonstrable therapeutic effect. The immunologic change induced by an allergen-specific DNA vaccine may not be the same as that of a protein-based allergen-specific vaccine. We have added a number of experiments in the legend of figure 4, as follows: Fig 4. Cytokine protein expression and the mRNA levels of splenocytes from each group of mice as determined by ELISA and qRT-PCR, respectively. (A) IL-13, (B) IL-4, (C) IL-10 and (D) IFN-γ. Cytokine release data are shown as the mean ± SEM from 3 independent experiments. qRT-PCR data are expressed as the mean fold increase ± SEM from 3 independent experiments. The statistical significance of differences among the groups was assessed by one-way analysis of variance with Dunnett’s t-test. *p<0.05; **p<0.01; NS, not statistically significant; SEM, standard error of the mean.”

Question 9. Figure 5 shows that eosinophils are higher in controls than any of the For t 2 vaccinated groups. in situ immunostaining in skin sections for CD4+ cells and intradermal IL-31. Patterns are referred to a “significantly” changed, without any possibility of statistical significance in this type of experiment. Furthermore, intensity of IL-31 staining is quantified in Fig 5C, with statistics applied inappropriately, but showing high significance. Text and statistics must be better aligned with the type of experiment shown.

Reply: To avoid confusion, we have rephrased the description of figure 5 on Page 13 as follows: “Fig 5A and 5B show that the total number of infiltrate cells, mainly the eosinophils, decreased significantly in the V50, V100, and V200 groups compared with the nonvaccinated and VO groups. IL-31 is known to be one of the key cytokines that induces itching and promotes scratching in mouse models of atopic dermatitis [18]. As shown in Fig 5C, expression of intradermal IL-31 was significantly decreased in all of the DNA vaccinated groups, including the VO group, compared with the nonvaccinated group. However, the reduction of IL-31 was more significant in the V50, V100, and V200 groups compared with the VO group.” 

Question 10. Figure 6 shows an experiment where groups of mice are sensitized with crude midge extract and then vaccinated before challenge with crude midge extract. Vaccination is shown to reduce scratching and IL-13 levels.

In order to have a more rigorous series of animal experiments, two additional experiments are needed, where E-rFor t 2 is used to sensitize and crude midge extract is used to challenge, either with or without vaccination, and also where crude midge extract is used to sensitize and E-For t 2 is used to challenge. These variations will help address some of the concerns about antigens that have been applied here.

Reply: The experiments shown in figure 6 are an attempt to reproduce a real-world scenario involving a human being bitten by midges (i.e., being exposed to all midge allergens after being bitten in the field). It is very unlikely that a human would be exposed to For t 2 protein only after being bitten by midges. We believe that a protocol involving a crude midge extract-challenge in which mice are sensitized to crude-extract is sufficient to represent a human response to midge allergy. 

Thank you again for reviewing our manuscript and for the reviewers’ valuable comments. We welcome your suggestions and advice. I look forward to hearing from you soon.

Sincerely,

Yi-Hsing Chen, MD, PhD.

Associate Professor of Medicine

Division of Allergy, Immunology and Rheumatology

Taichung Veterans General Hospital

Faculty of Medicine, National Yang-Ming University, 

1650 Taiwan Boulevard, Section 4, Taichung 40705, Taiwan

Tel. 886-4-23592525 ext. 3330

Fax 886-4-23592705

ysanne@vghtc.gov.tw

yhchen8@ym.edu.tw

---

## [Decision Letter · Decision Letter 1]

7 Apr 2020

Therapeutic DNA vaccine attenuates itching and allergic inflammation in mice with established biting midge allergy

PONE-D-20-00004R1

Dear Dr. Chen,

We are pleased to inform you that your manuscript has been judged scientifically suitable for publication and will be formally accepted for publication once it complies with all outstanding technical requirements.

With kind regards,

Junji Xing, Ph.D.

Academic Editor

PLOS ONE

Additional Editor Comments (optional):

Reviewers' comments:

Reviewer's Responses to Questions

**Comments to the Author**

1. If the authors have adequately addressed your comments raised in a previous round of review and you feel that this manuscript is now acceptable for publication, you may indicate that here to bypass the “Comments to the Author” section, enter your conflict of interest statement in the “Confidential to Editor” section, and submit your "Accept" recommendation.

Reviewer #1: All comments have been addressed

Reviewer #2: All comments have been addressed

2. Is the manuscript technically sound, and do the data support the conclusions?

Reviewer #1: Yes

Reviewer #2: Yes

3. Has the statistical analysis been performed appropriately and rigorously? 

Reviewer #1: Yes

Reviewer #2: Yes

4. Have the authors made all data underlying the findings in their manuscript fully available?

Reviewer #1: Yes

Reviewer #2: Yes

5. Is the manuscript presented in an intelligible fashion and written in standard English?

Reviewer #1: Yes

Reviewer #2: Yes

6. Review Comments to the Author

Reviewer #1: (No Response)

Reviewer #2: Authors have responded to all concerns and comments and have introduced better explanations of phenomena

7. PLOS authors have the option to publish the peer review history of their article (what does this mean?). If published, this will include your full peer review and any attached files.

Reviewer #1: No

Reviewer #2: No

---

## [Editor Report · Acceptance letter]

10 Apr 2020

PONE-D-20-00004R1 

Therapeutic DNA vaccine attenuates itching and allergic inflammation in mice with established biting midge allergy 

Dear Dr. Chen:

I am pleased to inform you that your manuscript has been deemed suitable for publication in PLOS ONE. Congratulations! Your manuscript is now with our production department. 

With kind regards,

on behalf of

Dr. Junji Xing 

Academic Editor

PLOS ONE